# Diabetic Retinopathy as a Risk Factor for Chronic Kidney Disease Progression: A Multicenter Case–Control Study in Taiwan

**DOI:** 10.3390/nu11030509

**Published:** 2019-02-27

**Authors:** Hsin-Ting Lin, Cai-Mei Zheng, Yun-Chun Wu, Yun-Hsiang Chang, Jiann-Torng Chen, Chang-Min Liang, Tian-Jong Chang, Jing-Quan Zheng, Ming-Cheng Tai, Yuh-Feng Lin

**Affiliations:** 1Department of Ophthalmology, Tri-Service General Hospital, National Defense Medical Center, Taipei 114, Taiwan; t72010@yahoo.com.tw (H.-T.L.); yun.siang@me.com (Y.-H.C.); jt66chen@gmail.com (J.-T.C.); doc30875@yahoo.com.tw (C.-M.L.); 2Graduate Institute of Medical Sciences, National Defense Medical Center, Taipei 114, Taiwan; 3Graduate Institute of Clinical Medicine, College of Medicine, Taipei Medical University, Taipei 110, Taiwan; 11044@s.tmu.edu.tw (C.-M.Z.); jingquan235@gmail.com (J.-Q.Z.); 4Department of Internal Medicine, School of Medicine, College of Medicine, Taipei Medical University, Taipei 110, Taiwan; 5Division of Nephrology, Department of Internal Medicine, Shuang Ho Hospital, Taipei Medical University, Taipei 235, Taiwan; 6Institute of Epidemiology and Preventive Medicine, College of Public Health, National Taiwan University, Taipei 100, Taiwan; luluchun@gmail.com; 7Graduate Institute of Aerospace and Undersea Medicine, National Defense Medical Center, Taipei 114, Taiwan; 8Performance Appraisal Section, Secretary Office, Shuang Ho Hospital, Taipei Medical University, Taipei 235, Taiwan; 10247@s.tmu.edu.tw; 9Department of Critical Care Medicine, Shuang Ho Hospital, Taipei Medical University, Taipei 235, Taiwan

**Keywords:** chronic kidney disease (CKD) cohort, diabetic retinopathy (DR), progression of CKD, estimated glomerular filtration rate, National Health Insurance Research Database

## Abstract

It has rarely been studied whether the presence and severity of diabetic retinopathy (DR) could influence the renal disease progression among all chronic kidney disease (CKD) diabetic patients. This study investigates the characteristics of diabetic patients, with different stages of chronic kidney disease (CKD), according to the occurrence of diabetic retinopathy and determines the influence of retinopathy in the deterioration of renal function. We conduct a multicenter, longitudinal cohort study based on the Epidemiology and Risk Factors Surveillance of the CKD project (2008–2013) and the National Health Insurance Research Database (NHIRD) (2001–2013). A total of 4050 diabetic patients with CKD, 20–85 years of age, from 14 hospitals and the community are included in this study. As compared to CKD patients without DR, CKD patients with DR have a lower baseline estimated glomerular filtration rate (eGFR) (39.17 ± 30.36 mL/min per 1.73 m^2^ vs. 54.38 ± 33.67 mL/min per 1.73 m^2^ ); poorer glycemic control (higher glycated hemoglobin (HbA1c) 7.85 ± 4.97 vs. 7.29 ± 4.02, *p* < 0.01); higher proteinuria (urine protein-to-creatinine ratio (UPCR )1.94 ± 2.96 g/dL vs. 0.91 ± 2.11 g/dL, *p* < 0.01); more anemia (Hb 11.22 ± 2.43 g/dL vs. 12.39 ± 3.85 g/dL, *p* < 0.01), and more hypoalbuminemia (3.88 ± 0.95 g/dL vs. 4.16 ± 1.74 g/dL, *p* < 0.01). Later stage (stage 3b–5) CKD patients with DR had significantly higher CKD progression compared with patients without DR (OR (odds ratio) 1.66 (1.36–2.02)). Patients with proliferative DR had significantly higher CKD progression events compared to patients with non-proliferative DR (OR 2.18 (1.71–2.78)). The presence and severity of DR is a risk factor for CKD progression among our Taiwanese CKD patients with diabetes. Prevention and early detection of DR are important and DR should be routinely screened as early as possible among diabetic CKD patients.

## 1. Introduction

Chronic kidney disease (CKD) is a serious global health problem with a high global prevalence (11%–13%) [1]. It represents a massive health cost because of its relationship with end-stage renal disease (ESRD) and dialysis. Taiwan has one of the highest rates of both CKD [2] and ESRD [3], globally. Diabetes mellitus (DM) is one of the most common causes of ESRD in in Taiwan [4], due to diabetic microvascular complications [5]. In patients with DM, chronic hyperglycemia is the most important factor related to microvascular complications [6]. The common pathological abnormalities within the retinal and glomerular microvascular beds are the progressive narrowing and eventual occlusion of vascular lumina within the retina [7,8] and glomerulus [9]. Although the presence of DR is considered the diagnosis of concomitant diabetic nephropathy (DN) [10,11], many studies suggest that the prevalence of non-diabetic renal diseases (NDRD), including primary glomerulonephritis, varies from 27% to 82.9% [12,13,14]. In other words, nearly 25%–50% of patients with DM have CKD that is not related to their diabetic conditions.

Clinically, both CKD and DR share common risk factors, such as aging, poor glycemic control, and systolic hypertension, etc. [15]. In these patients, similar histopathological lesions are noted within the glomerular and retinal vessels [16,17,18]. Hypertension, hypoalbuminemia, and proteinuria are independently associated with the progression of CKD in patients with DM [19]. The estimated glomerular filtration rate (eGFR), proteinuria, and the urine protein-to-creatinine ratio (UPCR) are used clinically as markers to evaluate renal function. Microalbuminuria is related to diabetic retinopathy (DR) and is considered a marker of retinopathy [20]. Similarly, many studies demonstrate that the presence of retinal microvascular abnormalities is associated with decline in renal function [21,22,23] and the absence of retinopathy in patients with better indicated renal prognoses, independent of blood pressure or metabolic control [23].

In a previous population-based study, diabetic retinopathy is found to be closely related to microalbuminuria and overt nephropathy among Korean patients with DM [24]. However, we still do not know the influence of the presence and severity of DR in CKD progression among various stages of CKD, from earlier to later stages, in Taiwanese patients with DM. Thus, in this population-based study, we evaluate the characteristics of chronic kidney disease (CKD) patients, with or without diabetic retinopathy (DR), and examine the relation between DR and its severity on the decline rate in the eGFR in stage 1–5 CKD patients.

## 2. Materials and Methods

### 2.1. Study Cohort

We conducted a multicenter, longitudinal cohort study from 14 medical centers and the community, collecting questionnaire and biochemistry data to establish the Epidemiology and Risk Factors Surveillance of the CKD database (2008–2013) by our core Laboratory and the Bureau of Health Promotion, Ministry of Health and Welfare, Taiwan. A total of 16,206 patients with and without CKD from 14 hospitals and the community, ranging from 20 to 85 years of age, were recruited in this study. End-stage renal disease (ESRD) was defined either as receiving maintenance dialysis or having a kidney transplant during this period. Patients with under 1-year of follow up, non-CKD, and missing or incomplete data were excluded. After exclusion, a total of 8537 participants who followed up for at least one year were included. We then linked the biochemical laboratory data to the National Health Insurance Research Database (NHIRD) from 2001 to 2013. During data analysis we further excluded 4487 patients without diabetes mellitus. Finally, 4050 patients with diabetes mellitus were included in the study analysis (Figure 1). Among these patients, 1481 had diabetic retinopathy and 2569 had no diabetic retinopathy. In those with DR, 963 had non-proliferative DR and 518 had proliferative DR (Figure 1). The patients were traced from the baseline date until the end of the study period (18 June 2013). The same laboratory criteria and protocols were applied across our study hospitals. We measured the CKD progression at the individual level by comparing eGFR and all of the creatinine levels were standardized. All of the participants were re-examined in the same hospital to prevent individual variations. The requirement for informed consent was exempted by the joint institutional review board before any data collection occurred. All study methods and recruitment processes followed joint institutional review board guidelines and the aforementioned governmental regulations.

### 2.2. Measurements and Variable Definitions

Data on patient demographics and health were collected using standardized questionnaires. These questionnaires included questions about age, sex, cigarette smoking, alcohol consumption, betel nut chewing, personal and family comorbid conditions, and medication use. Physical examinations were done especially on anthropometry, blood pressure (BP) and pulse rate (PR), and local and systemic examinations. Body height was measured by using a wall-mounted measuring tape (centimeters) and weight was measured by a digital scale in kilograms (SECA, model 782 2321009; Vogel & Halke, Hamburg, Germany). Diabetic retinopathy was diagnosed and staged clinically by dilated fundus examination and fluorescein angiography of the retina was used, if required. Staging of DR after the examination was done as follows: No apparent signs of DR was regarded as normal; microaneurysms, hard exudates, intraretinal hemorrhages, venous beading, or prominent intraretinal microvascular abnormality was regarded as early stage, non-proliferative DR (NPDR); and retinal or optic disk neovascularization, vitreous hemorrhage, or preretinal hemorrhage was regarded as late stage, proliferative DR (PDR) [25]. Proteinuria was determined by using the urine protein-to-creatinine ratio (UPCR). CKD stages were determined according to the Kidney Disease Outcomes Quality Initiative guidelines [26] using the estimated glomerular filtration rate (eGFR) and calculated by the Chronic Kidney Disease Epidemiology Collaboration equation as follows: eGFR (mL/min/1.73 m^2^) = 141 × min (SCr/ƙ, 1)α × max (serum creatinine/ƙ, 1) −1.209 × 0.993Age × 1.018 (if female) and × 1.159 (if black), where sCr denotes the serum creatinine level (mg/dL), ƙ = 0.7 (for women) and 0.9 (for men), α = −0.329 (for women) and −0.411(for men), min denotes the minimum of sCr/ƙ or 1, and max denotes the maximum of sCr/ƙ or 1 [27]. The various CKD stages include the following: Stage 1, eGFR ≥ 90 mL/min/1.73 m^2^ and the presence of any kidney damage (i.e., proteinuria dipsticks ≥ 1+, UPCR ≥ 150, or urine albumin-to-creatinine ratio (UACR) ≥ 30); stage 2, eGFR = 60–89 mL/min/1.73 m^2^ with any kidney damage (i.e., proteinuria dipsticks ≥ 1+, UPCR ≥ 150, or UACR ≥ 30); stage 3a, eGFR = 45–59 mL/min/1.73 m^2^; stage 3b, eGFR = 30–44 mL/min/1.73 m^2^; and stage 4, eGFR = 15–29 mL/min/1.73 m^2^; and stage 5, eGFR < 15 mL/min/1.73 m^2^ [28]. An average eGFR decline by more than 5 mL/min/1.73 m^2^ per year was defined as CKD progression [29].

### 2.3. Statistical Analysis

The odds ratio (OR) (95% confidence interval) of CKD was calculated for the overall, the non-proliferative DR, and the proliferative DR groups. We then explored the confounding and effect modification data in stratified analyses. After adjustments for all covariates, a multivariate logistic model with stepwise variable selection models was used to evaluate the relationship between DR and CKD progression. All statistical tests used the SAS statistical package (version 9.3; SAS Institute Inc., Cary, NC, USA). The value *p* < 0.05 was considered to be statistically significant.

### 2.4. Ethics Statement

This study was reviewed and approved by the institutional ethical committees of Taipei Medical University—Shuang Ho Hospital (TMU-JIRB 201204036), Tri-Service General Hospital (TSGHIRB100-05-197), Cardinal Tien Hospital (TMU-JIRB 201204035), Changhua Christian Hospital (CCHIRB 20405), Kaohsiung Medical University Chung-Ho Memorial Hospital (KMUHIRB 20120019), Kaohsiung Chang Gung Memorial Hospital (101-1096B), National Cheng Kung University Hospital (A-ER-101-117), and China Medical University Hospital (DMR101-IRB2-273(CR-1)). Written informed consent was obtained from all participants after a detailed explanation of the study protocol. Clinical and biological samples were collected by our trained surveyors and research assistants after patients’ informed consent was obtained.

## 3. Results

### 3.1. Demographic Characteristics of the Patients

Table 1 presents the basic characteristics of CKD patients with Diabetic Retinopathy (DR) and without Diabetic Retinopathy (Non-DR group). The DR group was younger than the non-DR group at baseline; the DR group was 64.95 ± 11.22 years (55.3% male) compared to 66.57 ± 12.11 years (57.49% male) in the non-DR group. Sex, BMI, and life-style habits (including exercise, smoking, alcohol consumption, and betel nut chewing) did not significantly differ between the two groups. The DR group had a significantly higher proportion of hypertension events (95.07% in DR group vs. 87.62% in non-DR group, *p* < 0.01) and blood pressure >130/85 mmHg (55.59% in DR group vs 49.64% in non-DR group, *p* < 0.01) than the non-DR group. There is significantly higher fasting glucose (AC glucose) 142.85 ± 64.63 mg/dL in the DR group vs. 129.09 ± 46.62 mg/dL in the non-DR group (*p* < 0.01). A total of 58.82% of the DR group vs. 43.96% of the non-DR group had higher glycated hemoglobin (HbA1c) ≧7% (*p* < 0.01). A significantly higher number of patients had UPCR > 1000 mg/dL (43.19% in the DR group vs. 21.08% in the non-DR group, *p* < 0.01). Up to 18.9% of the DR patients had severe UPCR > 3500 mg/dL, however, only 6.99% of non-DR patients had severe UPCR > 3500 mg/dL. The mean baseline estimated glomerular filtration rate (eGFR) was significantly lower, 39.17 ± 30.36 mL/min per 1.73 m^2^ in the DR group compared to 54.38 ± 33.67 mL/min per 1.73 m^2^ in the non-DR group. More CKD progression events were noted in the DR group than in the non-DR group (21.67% in the DR group vs. 13.62% in the non-DR group, *p* < 0.01). There was significantly lower hemoglobin (Hb) (11.22 ± 2.43 g/dL in the DR group vs. 12.39 ± 3.85 g/dL in the non-DR group (*p* < 0.01)), hematocrit (Hct) (32.84 ± 6.43% in DR group vs. 36.49 ± 6.48% in the non-DR group (*p* < 0.01)), and serum albumin (3.88 ± 0.95 g/dL in the DR group vs. 4.16 ± 1.74 g/dL in the non-DR group (*p* < 0.01)), respectively (Table 1).

### 3.2. Analysis of CKD Progression among the DR and Non-DR Groups

The proportion of CKD progression events was determined for all CKD stages, CKD stages 1–5, in the DR and non-DR groups. We further analyzed DR as a risk factor for CKD progression before and after adjustments for baseline eGFR, hypertension, HbA1c, and BMI (Table 2 and Table 3). We also executed the analysis for risk of CKD progression, according to DR severity for both non-proliferative DR and proliferative DR, within the DR group (Table 2 and Table 3).

### 3.3. Association between DR and CKD Progression in Different CKD Stages

Table 2 and Table 3 present the odds of CKD progression for various CKD stages, early CKD as stages 1–3a and late CKD as stages 3b–5, in the DR and non-DR groups. The study outcomes are represented as odds ratios (OR), by using the non-DR group as the reference to calculate the OR of CKD progression, in stages 1–3a and stages 3b–5, in both the DR and the non-DR groups. Significantly higher crude and adjusted OR of CKD progression were noted among overall and late CKD stages (Table 3) in the DR group than in the non-DR group. The crude OR of CKD progression events were 1.75 (1.48–2.07) * in all CKD stages (Table 2), 0.80 (0.54–1.19) in early CKD (stages 1–3a), and 1.66 (1.36–2.02) * in late CKD (stages 3b–5) in the DR group as compared to the non-DR group. After adjustment of baseline eGFR, hypertension, HbA1c, and BMI (model 3), CKD progression events are still significantly higher with 1.37 (1.08–1.73) * in all CKD stages (Table 2), 0.74 (0.44–1.24) in early CKD stages 1–3a and 1.47 (1.10–1.98) * in late CKD stages 3b–5 (Table 3).

### 3.4. Assoiciation between DR and CKD Progression According to Different DR Severity Status

Table 2 and Table 3 analyzes the odds ratio of CKD progression according to DR severity, overall DR, non-proliferative DR and proliferative DR. In these analyses, and the non-DR group was used as the reference group to calculate the OR of CKD progression for overall, proliferative and non-proliferative DR groups. The crude OR of CKD progression was significantly higher among overall, non-proliferative DR and proliferative DR groups; however, significantly higher adjusted OR (model 3) was noted only in overall and proliferative DR groups (Table 3). The crude OR (95% CIs) of CKD progression events were 1.75 (1.48–2.07) *, 1.57 (1.29–1.91) *, and 2.18 (1.71–2.78) *, for overall, non-proliferative DR, and proliferative DR, respectively (Table 2 and Table 3). After adjustments for baseline eGFR, hypertension, HbA1c and BMI (model 3), CKD progression remained significantly higher, with OR (95% CIs) of 1.37 (1.08–1.73) *, 1.25 (0.96–1.65), and 1.82 (1.31–2.54) *, for overall, non-proliferative DR, and proliferative DR, respectively (Table 2 and Table 3).

## 4. Discussion

This is a prospective cohort population study exploring the characteristics of a CKD cohort according to DR status. Subsequently, we determined the association between DR and CKD progression risk in different CKD stages. We then explored the risk of CKD progression according to DR severity. To our knowledge, this is the first prospective cohort study, linking to National Health Insurance Research Database, to evaluate the influence of DR as a risk factor for CKD progression among various stages of CKD according to DR severity in the Taiwanese population.

From baseline data, we observed that the CKD DR group was younger than CKD non-DR group in our CKD population. Other studies on patients with DM also find that DR patients are younger than those without DR [30,31]. Whether this observation between younger age and DR represents cause and effect is unclear, however, CKD patients should be evaluated for DR as early as possible after CKD diagnosis. In our CKD population, no significant differences were noted regarding sex and life-style habits, including exercise, smoking, or betel nut chewing. Our CKD DR patients had significantly worse glycemic control and higher blood pressure than the CKD non-DR group. Previous studies of diabetic patients with nephropathy also observe baseline characteristics similar to our patients in the DR group [31,32]. Previous studies find that moderate to severe CKD are at higher risk for retinopathy compared to milder CKD stages [33,34] and estimate patients with GFR <30 mL/min per 1.73 m^2^ have a three times higher risk for retinopathy [33]. Similarly, we noted poorer baseline eGFR and lower serum albumin levels with more severe proteinuria among our patients with DR, which might be explained by similar pathophysiological mechanisms underlying the development of microvascular damage in the retina and kidneys [22].

Significantly higher CKD progression events were noted in the DR group compared to the non-DR group (21.67% in the DR group vs. 13.62% in the non-DR group, *p* < 0.01) (Table 1). A previous prospective cohort study by Trevisan et.al. [23] also suggests that concomitant nephropathy and diabetic retinopathy is associated with a faster renal function deterioration in diabetic patients. An epidemiological study suggests that the narrowing of the retinal arterioles is associated with CKD, even in the absence of hypertension and diabetes [35]. The exact mechanism underlying the vasoconstriction of the retinal arterioles, which might be associated with renal dysfunction, remains unclear. A previous study of macroalbuminuric type 2 diabetic patients observes that those with retinopathy had Kimmenlstiel–Wilson nodules in kidney biopsies [11] and suggests that retinopathy is correlated with more advanced glomerular lesions. Furthermore, CKD progression events significantly increased in late CKD (stages 3b–5) compared to early CKD (stages 1–3a) (Table 3). This relationship persisted after adjustments for baseline eGFR, hypertension, HbA1c, and BMI (Table 3, model 3). The predictive value of DR in DN is still controversial because both are closely associated and interdependent microvascular complications of diabetes mellitus. A study finds that retinopathy is independently associated with a 10 year incidence of renal function impairment in patients with early onset type 1 diabetes [36]. The Reduction of Endpoints in NIDDM with the Angiotensin II Antagonist Losartan (RENAAL) study reveals similar results with worsening renal function and mortality in cases of retinopathy [32]. Similar results from a study on 1394 elderly CKD patients [21] notes that the presence of DR is significantly associated with progressive renal dysfunction, independent of diabetes or hypertension [21]. However, a recent report from the patients with advanced diabetic nephropathy and anemia (TREAT study) does not discover such correlation after adjustments for baseline eGFR or proteinuria [31]. Most studies include only patients with DM in later CKD stages, whereas our study cohort includes all CKD stages. Why DR is significantly related with CKD progression in later CKD stages is unknown. In the future, studies such as DIACORE [37], a prospective cohort study on 6000 patients with type 2 diabetes mellitus, involving a 10-year follow-up to elucidate incidence of microvascular and macrovascular complications, would further explain the relationship between retinopathy and renal function progression.

We note in our study that the DR severity is related with renal function decline in late CKD stages, independent of hypertension or diabetic status. Both proliferative and non-proliferative DR are significantly associated with CKD progression in unadjusted models, however, after adjustments for baseline eGFR, hypertension, HbA1c, and BMI, this relationship has been noted only in proliferative DR (Table 3, model 3). A multicenter, longitudinal cohort study on adults with chronic kidney disease (CKD) [38,39,40] reveals that a significant association exists between worse retinopathy scores and lower eGFR among diabetes patients. A previous meta-analysis revealed both proliferative and non-proliferative DR is useful in screening DN, with proliferative DR more specific than non-proliferative DR [41]. Community studies reveal that DR is closely associated with faster decline in the eGFR, regardless of the presence or absence of diabetes [21,22]. These findings reflect the hypothesis that both retinal and renal vascular beds share similar pathological and hemodynamic abnormalities, such as inflammatory processes [42,43], endothelial dysfunction [43] and reduced vascular reactivity [44,45] among CKD patients with diabetes. Common microscopic features in these vascular beds include thickening of the basement membrane and increased leakage [46]. Recent, genetic studies suggest that common inherited susceptibilities exist between retinopathy and CKD among patients with DM [47].

Our study has several limitations. In this study, since we used the prevalent CKD cohort, the effects of retinopathy on incident CKD were not determined. The personal identities of the study participants were linked to the national health care (NHI) databases. As the NHI database did not include the population that was not under medical health care it did not represent the whole population. However, in 2016, nearly 99.9 percent of Taiwanese population were under NHI coverage [48,49]. Thus, the effects of the missing population were negligible. Since the biochemical samples of the participants were collected individually by different hospitals and sent to the research center, some equipment variations might have influenced the results. However, all supervisors attempted to minimize all measurement errors. Furthermore, in this study, we did not determine the influence of DR on mortality outcomes among CKD patients and this information might be important in future studies.

## 5. Conclusions

In conclusion, the presence and severity of retinopathy intimately associates with CKD progression among all diabetes patients with chronic kidney diseases. Routine follow-up and management of ocular and retinal disorders in CKD patients with diabetes would be important for aggressive management of diabetic retinopathy and prevention of CKD progression among these patients.

## Figures and Tables

**Figure 1 nutrients-11-00509-f001:**
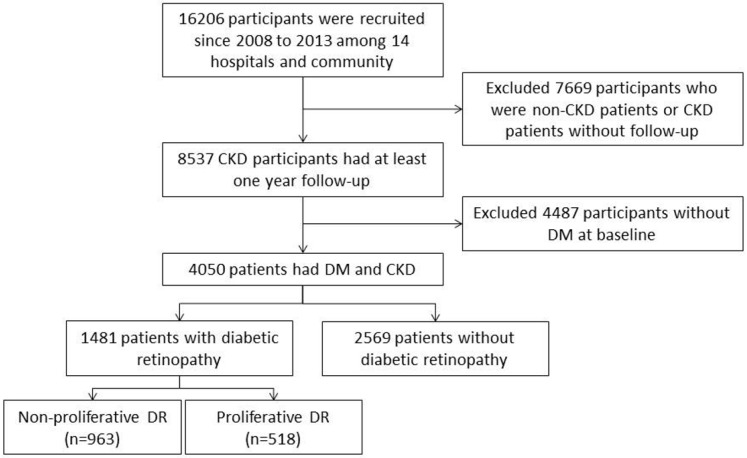
Recruitment process flow chart. The chronic kidney disease (CKD) cases were recruited from the nephrology outpatient clinics in each hospital and from the community. DM, diabetes mellitus; DR, diabetic retinopathy.

**Table 1 nutrients-11-00509-t001:** Basic characteristics of chronic kidney disease (CKD) patients in the group with diabetic retinopathy (DR) or the group without diabetic retinopathy (Non-DR).

Variables	Diabetic Retinopathy *n* = 1481 (%)		Without Diabetic Retinopathy *n* = 2569 (%)		*p*-Value *
Age, mean (SD), y	64.95 ± 11.22		66.57 ± 12.11		<0.01
Male Sex	819	(55.30)	1477	(57.49)	0.18
BMI, mean (SD), kg/m^2^	25.90 ± 4.09		26.02 ± 4.30		0.41
Waist, mean (SD), cm	90.03 ± 10.80		90.09 ± 11.65		0.88
Exercise	487	(34.71)	825	(34.01)	0.66
Smoking	385	(26.00)	704	(27.40)	0.33
Alcohol	125	(8.44)	268	(10.43)	0.04
Betel Nut	102	(6.89)	166	(6.46)	0.60
CKD progression	321	(21.67)	350	(13.62)	<0.01
Hypertension	1408	(95.07)	2251	(87.62)	<0.01
Blood pressure					<0.01
Less than 130/85	445	(44.41)	828	(50.36)	
Over 130/85	557	(55.59)	816	(49.64)	
HbA1C, mean (SD), %	7.85 ± 4.97		7.29 ± 4.02		<0.01
HbA1C (<7%)	460	(41.18)	840	(56.04)	<0.01
HbA1C (≥7%)	657	(58.82)	659	(43.96)	
Urine protein-to-creatinine ratio (UPCR), mean (SD), g/dL	1.94 ± 2.96		0.91 ± 2.11		<0.01
UPCR (<150)	241	(21.16)	871	(43.21)	<0.01
UPCR (150–1000)	406	(35.65)	720	(35.71)	
UPCR (1000–3500)	276	(24.23)	284	(14.09)	
UPCR (>3500)	216	(18.96)	141	(6.99)	
Baseline estimated glomerular filtration rate (eGFR), mean (SD), mL/min/1.73 m^2^	39.17 ± 30.36		54.38 ± 33.67		<0.01
Cholesterol, mean (SD), mg/dL	183.41 ± 50.19		183.15 ± 43.57		0.88
Triglyceride, mean (SD), mg/dL	159.25 ± 104.13		159.73 ± 115.46		0.90
Uric Acid, mean (SD), mg/dL	8.56 ± 41.07		10.09 ± 74.78		0.44
Glucose (AC), mean (SD), mg/dL	142.85 ± 64.63		129.09 ± 46.62		<0.01
Na, mean (SD), mmol/L	139.79 ± 40.07		139.16 ± 5.04		0.64
K, mean (SD), mmol/L	4.66 ± 4.26		4.55 ± 4.04		0.54
Ca, mean (SD), mmol/L	8.87 ± 0.92		9.10 ± 2.30		<0.01
P, mean (SD), mg/dL	4.30 ± 1.11		4.11 ± 4.59		0.17
Hb, mean (SD), g/dL	11.22 ± 2.43		12.39 ± 3.85		<0.01
Hct, mean (SD), %	32.84 ± 6.43		36.49 ± 6.48		<0.01
Albumin, mean (SD), g/dL	3.88 ± 0.95		4.16 ± 1.74		<0.01

* Categorical variables: Chi-squared test; continuous variables: Independent *t* test.

**Table 2 nutrients-11-00509-t002:** The association between diabetic retinopathy and risk of CKD progression.

Variables	Without Diabetic Retinopathy	Diabetic Retinopathy
	OR (95% CI)	OR (95% CI)
Univariate Model	1.00	1.75 (1.48–2.07) *
Multivariate Model 1	1.00	1.28 (1.07–1.52) *
Multivariate Model 2	1.00	1.32 (1.06–1.65) *
Multivariate Model 3	1.00	1.37 (1.08–1.73) *

Model 1 was adjusted by baseline eGFR and hypertension. Model 2 was adjusted by baseline eGFR, hypertension, and HbA1c. Model 3 was adjusted by baseline eGFR, hypertension, HbA1c, and BMI. * *p* < 0.05.

**Table 3 nutrients-11-00509-t003:** The association between diabetic retinopathy and risk of CKD progression by CKD stage and severity of diabetic retinopathy.

Variables	Without Diabetic Retinopathy	With Diabetic Retinopathy
	OR (95% CI)	OR (95% CI)
CKD stage		
Stage 1–3a		
Univariate Model	1.00	0.80 (0.54–1.19)
Multivariate Model 1	1.00	0.80 (0.54–1.19)
Multivariate Model 2	1.00	0.76 (0.47–1.25)
Multivariate Model 3	1.00	0.74 (0.44–1.24)
Stage 3b–5		
Univariate Model	1.00	1.66 (1.36–2.02) *
Multivariate Model 1	1.00	1.32 (1.06–1.63) *
Multivariate Model 2	1.00	1.40 (1.07–1.83) *
Multivariate Model 3	1.00	1.47 (1.10–1.98) *
Severity of DR		
Non-proliferative DR		
Univariate Model	1.00	1.57 (1.29–1.91) *
Multivariate Model 1	1.00	1.25 (1.02–1.53) *
Multivariate Model 2	1.00	1.26 (0.98–1.62)
Multivariate Model 3	1.00	1.25 (0.96–1.65)
Proliferative DR		
Univariate Model	1.00	2.18 (1.71–2.78) *
Multivariate Model 1	1.00	1.46 (1.13–1.88) *
Multivariate Model 2	1.00	1.64 (1.21–2.22) *
Multivariate Model 3	1.00	1.82 (1.31–2.54) *

Model 1 was adjusted by baseline eGFR and hypertension. Model 2 was adjusted by baseline eGFR, hypertension, and HbA1c. Model 3 was adjusted by baseline eGFR, hypertension, HbA1c, and BMI. * *p* < 0.05.

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
