# Peer review of "Diabetic Retinopathy as a Risk Factor for Chronic Kidney Disease Progression: A Multicenter Case–Control Study in Taiwan"

_nutrients, 2019, doi:10.3390/nu11030509_

Round 1

Reviewer 1 Report

Background: Background needs lots of proof-reading and editing.

The rationale for studying the association of diabetic retinopathy with CKD was not justified correctly. Although, it is really interesting as authors correctly described the shared pathways between diabetic retinopathy and chronic kidney diseases, the epidemiological link and causal link was not described properly.

Methods:

Explain in little more detail about how the diagnosis of Diabetic retinopathy assessed.

Results:

Table-1: Age appeared twice. Also, provide triglycerides in median (inter-quartile range)

Section 3.3 and 3.4. Change correlation to the association.

Discussion: Please avoid using strong statements like "proved". It appeared in 3-4 instances.

Please proof-read wherever necessary for clarity.

Conclusions: Please replace "predict" with "association" as the study is cross-sectional.

Author Response

Response to Reviewer 1 Comments

Question 1: Background: Background needs lots of proof-reading and editing. 

Answer 1: Thanks for suggestion. We did proof-reading and editing accordingly.

Question 2: The rationale for studying the association of diabetic retinopathy with CKD was not justified correctly. Although, it is really interesting as authors correctly described the shared pathways between diabetic retinopathy and chronic kidney diseases, the epidemiological link and causal link was not described properly.

Answer 2: Thanks for your great comment. We add the rationale and link between DR and CKD in the background as your suggestion (Page 2, Line 86-87).

Question 3: Methods: Explain in little more detail about how the diagnosis of Diabetic retinopathy assessed.

Answer 3: We appreciate the suggestion. The assessment of DR was described in Page 3, Line 191-196.

Question 4: Results: Table-1: Age appeared twice. Also, provide triglycerides in median (inter-quartile range) 

Answer 4: Thanks for your suggestions. We make appropriate corrections as your suggestion. The median level of triglyceride in our study patients was 134mg/dL.

Question 5: Section 3.3 and 3.4. Change correlation to the association.

Answer 5: Changes were done as suggestion.

Question 6: Discussion: Please avoid using strong statements like "proved". It appeared in 3-4 instances. 

Answer 6: Thanks for your suggestions. We make appropriate corrections as your suggestion.

Question 7: Please proof-read wherever necessary for clarity.

Answer 7: Thanks for your suggestions. We made a professional proof-read as your suggestion.

Question 8: Conclusions: Please replace "predict" with "association" as the study is cross-sectional.

Answer 7: Thanks for the comment. We replace "predict" with "associates" (Page 8, Line 585).

Reviewer 2 Report

Abstract: Generally clear and succinct. Highlights the key aim and finding from the study. DR is used as an abbreviation before it is defined. It is also unabbreviated in the keywords list yet used throughout.

Introduction:

Brief but outlines the basis for the study.

Line 49: Should read countries rather than country

Line 50: Poor wording/English (I stopped correcting each language mistake after this point, they are numerous throughout)

Materials and Methods:

Good sample size and range of hospitals involved.

Could avoid the use of etc. in descriptions, rather detail in full.

More information is needed in terms of the stats analysis section. Specifics around adjustments made for factors such as sig. diff. age difference in groups. Description is a little vague.

Results:

Line 145: AC has not been defined (AC glucose)

More detail needed around criteria for some lifestyle variables. Smoking/exercise/alcohol…. How are these being quantified? Is it a yes/no or a sliding scale? If a sliding scale what are the cut offs?

No reported data loss or mortality during follow up

Be consistent in the number of decimal places you are reporting P-values to. In both text and tables.

Discussion:

This section presents a sound evaluation of the key findings of the project and draws on findings from appropriate literature in the field.

Conclusions drawn are sound.

English language needs improving throughout

Author Response

Response to Reviewer 2 Comments

Question 1: Abstract: Generally clear and succinct. Highlights the key aim and finding from the study. DR is used as an abbreviation before it is defined. It is also unabbreviated in the keywords list yet used throughout.

Answer 1: Thanks for your great comment. We unabbreviated the word DR ahead in abstract as your suggestion.

Question 2: Introduction: Brief but outlines the basis for the study.

Answer 2: Thanks for your appreciation.

Question 3: Line 49: Should read countries rather than country

Answer 3: Thanks for your great suggestion. We corrected accordingly.

Question 4: Line 50: Poor wording/English (I stopped correcting each language mistake after this point, they are numerous throughout)

Answer 4: Thanks for your suggestions. We made a professional proof-read of our revised manuscript.

Question 5: Materials and Methods: Good sample size and range of hospitals involved. Could avoid the use of etc. in descriptions, rather detail in full.

Answer 5: Thanks for your great comment. We omit the word etc. as your suggestion.

Question 6: More information is needed in terms of the stats analysis section. Specifics around adjustments made for factors such as sig. diff. age difference in groups. Description is a little vague.

Answer 6: We used stepwise selection analysis to analyze the adjusted variables. These variables included eGFR, which includes age adjustment in its formula eGFR(mL/min/1.73 m2) =141 × min (SCr/ƙ, 1)α × max (serum creatinine/ƙ, 1)1.209 × 0.993Age × 1.018 (if female) and × 1.159 (if black), we didn’t specifically adjust age again to avoid overcorrection.

Question 7: Results: Line 145: AC has not been defined (AC glucose).

Answer 7: Thanks for your great suggestion. We corrected accordingly. It was described in Page 8, Line 253.

More detail needed around criteria for some lifestyle variables. Smoking/exercise/alcohol…. How are these being quantified? Is it a yes/no or a sliding scale? If a sliding scale what are the cut offs?

Answer 7: Life style behaviors of the participants were asked using the standard questionnaires at the time when they entered the study. The definition of smoking had been defined as participants had smoked more than 100 cigarettes (5 packs) in the past and smoking within recent one month. Alcohol consumption had been defined as having continuous drinking wine, more than 100ml/day for 3 months or liquor, more than 50ml/day for 3 months in the past and drinking within recent one month. Chewing betel nut had been defined as continuous chewing betel nut for more than 3 months with more than 5 betel nut/day, 3 days/week in the past and chewing betel nut within recent one month. Exercise had been defined as continuous exercise for more than 30 minutes/day, three times/week, continuous exercise for more than 3 months in the past and exercise within recent one month.

No reported data loss or mortality during follow up.

Answer 7: Thanks you for your great suggestion. In this study, we have excluded 7669 participants who were non-CKD patients or CKD patients with follow up less than one year including mortality cases (as shown in Figure 1).

Be consistent in the number of decimal places you are reporting P-values to. In both text and tables.

Answer 7: Thanks for your great suggestion. We corrected accordingly.

Question 8: Discussion: This section presents a sound evaluation of the key findings of the project and draws on findings from appropriate literature in the field. Conclusions drawn are sound.

Answer 8: Thanks for your great comment.

Question 9: English language needs improving throughout.

Answer 9: Thanks for your great comment. We did appropriate English editing as your suggestion.
